# Smooth Primal-Dual Coordinate Descent Algorithms for Nonsmooth Convex Optimization

**Ahmet Alacaoglu[1]**    **Quoc Tran-Dinh[2]**    **Olivier Fercoq[3]**    **Volkan Cevher[1]**

[1]Laboratory for Information and Inference Systems (LIONS), EPFL, Lausanne, Switzerland
`{ahmet.alacaoglu, volkan.cevher}@epfl.ch`
[2] Department of Statistics and Operations Research, UNC-Chapel Hill, NC, USA
`quoctd@email.unc.edu`
[3] LTCI, Télécom ParisTech, Université Paris-Saclay, Paris, France
`olivier.fercoq@telecom-paristech.fr`

## Abstract

We propose a new randomized coordinate descent method for a convex optimization template with broad applications. Our analysis relies on a novel combination of four ideas applied to the primal-dual gap function: smoothing, acceleration, homotopy, and coordinate descent with non-uniform sampling. As a result, our method features the first convergence rate guarantees among the coordinate descent methods, that are the best-known under a variety of common structure assumptions on the template. We provide numerical evidence to support the theoretical results with a comparison to state-of-the-art algorithms.

## 1 Introduction

We develop randomized coordinate descent methods to solve the following composite convex problem:

$$F^{\star} = \min_{x \in \mathbb{R}^p} \{ F(x) = f(x) + g(x) + h(Ax) \}, \qquad (1)$$

where $f : \mathbb{R}^p \to \mathbb{R}$, $g : \mathbb{R}^p \to \mathbb{R} \cup \{+\infty\}$, and $h : \mathbb{R}^m \to \mathbb{R} \cup \{+\infty\}$ are proper, closed and convex functions, $A \in \mathbb{R}^{m \times p}$ is a given matrix. The optimization template (1) covers many important applications including support vector machines, sparse model selection, logistic regression, etc. It is also convenient to formulate generic constrained convex problems by choosing an appropriate $h$.

Within convex optimization, coordinate descent methods have recently become increasingly popular in the literature [1–6]. These methods are particularly well-suited to solve huge-scale problems arising from machine learning applications where matrix-vector operations are prohibitive [1].

To our knowledge, there is no coordinate descent method for the general three-composite form (1) within our structure assumptions studied here that has rigorous convergence guarantees. Our paper specifically fills this gap. For such a theoretical development, coordinate descent algorithms require specific assumptions on the convex optimization problems [1, 4, 6]. As a result, to rigorously handle the three-composite case, we assume that $(i)$ $f$ is smooth, $(ii)$ $g$ is non-smooth but decomposable (each component has an "efficiently computable" proximal operator), and $(iii)$ $h$ is non-smooth.

**Our approach:**    In a nutshell, we generalize [4, 7] to the three composite case (1). For this purpose, we combine several classical and contemporary ideas: We exploit the smoothing technique in [8], the efficient implementation technique in [4, 14], the homotopy strategy in [9], and the nonuniform coordinate selection rule in [7] in our algorithm, to achieve the best known complexity estimate for the template.

Surprisingly, the combination of these ideas is achieved in a very natural and elementary primal-dual gap-based framework. However, the extension is indeed not trivial since it requires to deal with the composition of a non-smooth function $h$ and a linear operator $A$.

While our work has connections to the methods developed in [7, 10, 11], it is rather distinct. First, we consider a more general problem (1) than the one in [4, 7, 10]. Second, our method relies on Nesterov's accelerated scheme rather than a primal-dual method as in [11]. Moreover, we obtain the first rigorous convergence rate guarantees as opposed to [11]. In addition, we allow using any sampling distribution for choosing the coordinates.

**Our contributions:** We propose a new smooth primal-dual randomized coordinate descent method for solving (1) where $f$ is smooth, $g$ is nonsmooth, separable and has a block-wise proximal operator, and $h$ is a general nonsmooth function. Under such a structure, we show that our algorithm achieves the best known $\mathcal{O}(n/k)$ convergence rate, where $k$ is the iteration count and to our knowledge, this is the first time that this convergence rate is proven for a coordinate descent algorithm.

We instantiate our algorithm to solve special cases of (1) including the case $g = 0$ and constrained problems. We analyze the convergence rate guarantees of these variants individually and discuss the choices of sampling distributions.

Exploiting the strategy in [4, 14], our algorithm can be implemented in parallel by breaking up the full vector updates. We also provide a restart strategy to enhance practical performance.

**Paper organization:** We review some preliminary results in Section 2. The main contribution of this paper is in Section 3 with the main algorithm and its convergence guarantee. We also present special cases of the proposed algorithm. Section 4 provides numerical evidence to illustrate the performance of our algorithms in comparison to existing methods. The proofs are deferred to the supplementary document.

## 2  Preliminaries

**Notation:** Let $[n] := \{1, 2, \cdots, n\}$ be the set of $n$ positive integer indices. Let us decompose the variable vector $x$ into $n$-blocks denoted by $x_i$ as $x = [x_1; x_2; \cdots; x_n]$ such that each block $x_i$ has the size $p_i \geq 1$ with $\sum_{i=1}^n p_i = p$. We also decompose the identity matrix $\mathbb{I}_p$ of $\mathbb{R}^p$ into $n$ block as $\mathbb{I}_p = [U_1, U_2, \cdots, U_n]$, where $U_i \in \mathbb{R}^{p \times p_i}$ has $p_i$ unit vectors. In this case, any vector $x \in \mathbb{R}^p$ can be written as $x = \sum_{i=1}^n U_i x_i$, and each block becomes $x_i = U_i^\top x$ for $i \in [n]$. We define the partial gradients as $\nabla_i f(x) = U_i^\top \nabla f(x)$ for $i \in [n]$. For a convex function $f$, we use $\mathrm{dom}\,(f)$ to denote its domain, $f^*(x) := \sup_u \{u^\top x - f(u)\}$ to denote its Fenchel conjugate, and $\mathrm{prox}_f(x) := \arg\min_u \{f(u) + (1/2)\|u - x\|^2\}$ to denote its proximal operator. For a convex set $\mathcal{X}$, $\delta_{\mathcal{X}}(\cdot)$ denotes its indicator function. We also need the following weighted norms:

$$\|x_i\|_{(i)}^2 = \langle H_i x_i, x_i \rangle, \qquad (\|y_i\|_{(i)}^*)^2 = \langle H_i^{-1} y_i, y_i \rangle,$$
$$\|x\|_{[\alpha]}^2 = \sum_{i=1}^n L_i^\alpha \|x_i\|_{(i)}^2, \qquad (\|y\|_{[\alpha]}^*)^2 = \sum_{i=1}^n L_i^{-\alpha}(\|y_i\|_{(i)}^*)^2. \tag{2}$$

Here, $H_i \in \mathbb{R}^{p_i \times p_i}$ is a symmetric positive definite matrix, and $L_i \in (0, \infty)$ for $i \in [n]$ and $\alpha > 0$. In addition, we use $\|\cdot\|$ to denote $\|\cdot\|_2$.

**Formal assumptions on the template:** We require the following assumptions to tackle (1):

**Assumption 1.** *The functions $f$, $g$ and $h$ are all proper, closed and convex. Moreover, they satisfy*

(a) *The partial derivative $\nabla_i f(\cdot)$ of $f$ is Lipschitz continuous with the Lipschitz constant $\hat{L}_i \in [0, +\infty)$, i.e., $\|\nabla_i f(x + U_i d_i) - \nabla_i f(x)\|_{(i)}^* \leq \hat{L}_i \|d_i\|_{(i)}$ for all $x \in \mathbb{R}^p, d_i \in \mathbb{R}^{p_i}$.*

(b) *The function $g$ is separable, which has the following form $g(x) = \sum_{i=1}^n g_i(x_i)$.*

(c) *One of the following assumptions for $h$ holds for Subsections 3.3 and 3.4, respectively:*
   i. *$h$ is Lipschitz continuous which is equivalent to the boundedness of $\mathrm{dom}\,(h^*)$.*
   ii. *$h$ is the indicator function for an equality constraint, i.e., $h(Ax) := \delta_{\{c\}}(Ax)$.*

Now, we briefly describe the main techniques used in this paper.

**Acceleration:** Acceleration techniques in convex optimization date back to the seminal work of Nesterov in [13], and is one of standard techniques in convex optimization. We exploit such a scheme to achieve the best known $\mathcal{O}(1/k)$ rate for the nonsmooth template (1).

**Nonuniform distribution:** We assume that $\xi$ is a random index on $[n]$ associated with a probability distribution $q = (q_1, \cdots, q_n)^\top$ such that

$$\mathbb{P}\{\xi = i\} = q_i > 0, \ \ i \in [n], \ \ \text{and} \ \ \sum_{i=1}^n q_i = 1. \tag{3}$$

When $q_i = \frac{1}{n}$ for all $i \in [n]$, we obtain the uniform distribution. Let $i_0, i_1, \cdots, i_k$ be i.i.d. realizations of the random index $\xi$ after $k$ iteration. We define $\mathcal{F}_{k+1} = \sigma(i_0, i_1, \cdots, i_k)$ as the $\sigma$-field generated by these realizations.

**Smoothing techniques:** We can write the convex function $h(u) = \sup_y \{\langle u, y\rangle - h^*(y)\}$ using its Fenchel conjugate $h^*$. Since $h$ in (1) is convex but possibly nonsmooth, we smooth $h$ as

$$h_\beta(u) := \max_{y \in \mathbb{R}^m} \left\{ \langle u, y\rangle - h^*(y) - \frac{\beta}{2}\|y - \dot{y}\|^2 \right\}, \tag{4}$$

where $\dot{y} \in \mathbb{R}^m$ is given and $\beta > 0$ is the smoothness parameter. Moreover, the quadratic function $b(y, \dot{y}) = \frac{1}{2}\|y - \dot{y}\|^2$ is defined based on a given norm in $\mathbb{R}^m$. Let us denote by $y_\beta^*(u)$, the unique solution of this concave maximization problem in (4), i.e.:

$$y_\beta^*(u) := \arg\max_{y \in \mathbb{R}^m} \left\{ \langle u, y\rangle - h^*(y) - \frac{\beta}{2}\|y - \dot{y}\|^2 \right\} = \mathrm{prox}_{\beta^{-1}h^*}\left(\dot{y} + \beta^{-1}u\right), \tag{5}$$

where $\mathrm{prox}_{h^*}$ is the proximal operator of $h^*$. If we assume that $h$ is Lipschitz continuous, or equivalently that $\mathrm{dom}\,(h^*)$ is bounded, then it holds that

$$h_\beta(u) \le h(u) \le h_\beta(u) + \frac{\beta D_{h^*}^2}{2}, \quad \text{where } D_{h^*} := \max_{y \in \mathrm{dom}(h^*)} \|y - \dot{y}\| < +\infty. \tag{6}$$

Let us define a new smoothed function $\psi_\beta(x) := f(x) + h_\beta(Ax)$. Then, $\psi_\beta$ is differentiable, and its block partial gradient

$$\nabla_i \psi_\beta(x) = \nabla_i f(x) + A_i^\top y_\beta^*(Ax) \tag{7}$$

is also Lipschitz continuous with the Lipschitz constant $L_i(\beta) := \hat{L}_i + \frac{\|A_i\|^2}{\beta}$, where $\hat{L}_i$ is given in Assumption 1, and $A_i \in \mathbb{R}^{m \times p_i}$ is the $i$-th block of $A$.

**Homotopy:** In smoothing-based methods, the choice of the smoothness parameter is critical. This choice may require the knowledge of the desired accuracy, number of maximum iterations or the diameters of the primal and/or dual domains as in [8]. In order to make this choice flexible and our method applicable to the constrained problems, we employ a homotopy strategy developed in [9] for deterministic algorithms, to gradually update the smoothness parameter while making sure that it converges to 0.

## 3  Smooth primal-dual randomized coordinate descent

In this section, we develop a smoothing primal-dual method to solve (1). Or approach is to combine the four key techniques mentioned above: smoothing, acceleration, homotopy, and randomized coordinate descent. Similar to [7] we allow to use arbitrary nonuniform distribution, which may allow to design a good distribution that captures the underlying structure of specific problems.

### 3.1  The algorithm
Algorithm 1 below smooths, accelerates, and randomizes the coordinate descent method.

---

**Algorithm 1.** SMooth, Accelerate, Randomize The Coordinate Descent (SMART-CD)

---

**Input:** Choose $\beta_1 > 0$ and $\alpha \in [0, 1]$ as two input parameters. Choose $x^0 \in \mathbb{R}^p$.

1   Set $B_i^0 := \hat{L}_i + \frac{\|A_i\|^2}{\beta_1}$ for $i \in [n]$. Compute $S_\alpha := \sum_{i=1}^n (B_i^0)^\alpha$ and $q_i := \frac{(B_i^0)^\alpha}{S_\alpha}$ for all $i \in [n]$.

2   Set $\tau_0 := \min\{q_i \mid 1 \le i \le n\} \in (0, 1]$ for $i \in [n]$. Set $\bar{x}^0 = \tilde{x}^0 := x^0$.

3   **for** $k \leftarrow 0, 1, \cdots, k_{\max}$ **do**

4       Update $\hat{x}^k := (1 - \tau_k)\bar{x}^k + \tau_k \tilde{x}^k$ and compute $\hat{u}^k := A\hat{x}^k$.

5       Compute the dual step $y_k^* := y_{\beta_{k+1}}^*(\hat{u}^k) = \mathrm{prox}_{\beta_{k+1}^{-1}h^*}\left(\dot{y} + \beta_{k+1}^{-1}\hat{u}^k\right)$.

6       Select a block coordinate $i_k \in [n]$ according to the probability distribution $q$.

7       Set $\tilde{x}^{k+1} := \tilde{x}^k$, and compute the primal $i_k$-block coordinate:

$$\tilde{x}_{i_k}^{k+1} := \mathop{\mathrm{argmin}}_{x_{i_k} \in \mathbb{R}^{p_{i_k}}} \left\{ \langle \nabla_{i_k} f(\hat{x}^k) + A_{i_k}^\top y_k^*, x_{i_k} - \hat{x}_{i_k}^k\rangle + g_{i_k}(x_{i_k}) + \frac{\tau_k B_{i_k}^k}{2\tau_0}\|x_{i_k} - \tilde{x}_{i_k}^k\|_{(i_k)}^2 \right\}.$$

8       Update $\bar{x}^{k+1} := \hat{x}^k + \frac{\tau_k}{\tau_0}(\tilde{x}^{k+1} - \tilde{x}^k)$.

9       Compute $\tau_{k+1} \in (0, 1)$ as the unique positive root of $\tau^3 + \tau^2 + \tau_k^2\tau - \tau_k^2 = 0$.

10      Update $\beta_{k+2} := \frac{\beta_{k+1}}{1+\tau_{k+1}}$ and $B_i^{k+1} := \hat{L}_i + \frac{\|A_i\|^2}{\beta_{k+2}}$ for $i \in [n]$.

11  **end for**

---

From the update $\bar{x}^k := \hat{x}^{k-1} + \frac{\tau_{k-1}}{\tau_0}(\tilde{x}^k - \tilde{x}^{k-1})$ and $\hat{x}^k := (1 - \tau_k)\bar{x}^k + \tau_k\tilde{x}^k$, it directly follows that $\hat{x}^k := (1 - \tau_k)\big(\hat{x}^{k-1} + \frac{\tau_{k-1}}{\tau_0}(\tilde{x}^k - \tilde{x}^{k-1})\big) + \tau_k\tilde{x}^k$. Therefore, it is possible to implement the algorithm without forming $\bar{x}^k$.

## 3.2 Efficient implementation

While the basic variant in Algorithm 1 requires full vector updates at each iteration, we exploit the idea in [4, 14] and show that we can partially update these vectors in a more efficient manner.

---

**Algorithm 2.** Efficient SMART-CD

---

**Input:** Choose a parameter $\beta_1 > 0$ and $\alpha \in [0, 1]$ as two input parameters. Choose $x^0 \in \mathbb{R}^p$.

1  Set $B_i^0 := \hat{L}_i + \frac{\|A_i\|^2}{\beta_1}$ for $i \in [n]$. Compute $S_\alpha := \sum_{i=1}^n (B_i^0)^\alpha$ and $q_i := \frac{(B_i^0)^\alpha}{S_\alpha}$ for all $i \in [n]$.

2  Set $\tau_0 := \min\{q_i \mid 1 \le i \le n\} \in (0, 1]$ for $i \in [n]$ and $c_0 = (1 - \tau_0)$. Set $u^0 = \tilde{z}^0 := x^0$.

3  **for** $k \leftarrow 0, 1, \cdots, k_{\max}$ **do**

4    Compute the dual step $y_{\beta_{k+1}}^*(c_k Au^k + A\tilde{z}^k) := \text{prox}_{\beta_{k+1}^{-1}h^*}\big(\dot{y} + \beta_{k+1}^{-1}(c_k Au^k + A\tilde{z}^k)\big)$.

5    Select a block coordinate $i_k \in [n]$ according to the probability distribution $q$.

6    Let $\nabla_i^k := \nabla_{i_k} f(c_k u^k + \tilde{z}^k) + A_{i_k}^\top y_{\beta_{k+1}}^*(c_k Au^k + A\tilde{z}^k)$. Compute

$$t_{i_k}^{k+1} := \arg\min_{t \in \mathbb{R}^{p_{i_k}}} \left\{ \langle \nabla_i^k, t \rangle + g_{i_k}(t + \tilde{z}_{i_k}^k) + \frac{\tau_k B_{i_k}^k}{2\tau_0}\|t\|_{(i_k)}^2 \right\}.$$

7    Update $\tilde{z}_{i_k}^{k+1} := \tilde{z}_{i_k}^k + t_{i_k}^{k+1}$.

8    Update $u_{i_k}^{k+1} := u_{i_k}^k - \frac{1 - \tau_k/\tau_0}{c_k}t_{i_k}^{k+1}$.

9    Compute $\tau_{k+1} \in (0, 1)$ as the unique positive root of $\tau^3 + \tau^2 + \tau_k^2\tau - \tau_k^2 = 0$.

10   Update $\beta_{k+2} := \frac{\beta_{k+1}}{1 + \tau_{k+1}}$ and $B_i^{k+1} := \hat{L}_i + \frac{\|A_i\|^2}{\beta_{k+2}}$ for $i \in [n]$.

11  **end for**

---

We present the following result which shows the equivalence between Algorithm 1 and Algorithm 2, the proof of which can be found in the supplementary document.

**Proposition 3.1.** *Let $c_k = \prod_{l=0}^k (1 - \tau_l)$, $\hat{z}^k = c_k u^k + \tilde{z}^k$ and $\bar{z}^k = c_{k-1} u^k + \tilde{z}^k$. Then, $\tilde{x}^k = \tilde{z}^k$, $\hat{x}^k = \hat{z}^k$ and $\bar{x}^k = \bar{z}^k$, for all $k \ge 0$, where $\tilde{x}^k, \hat{x}^k$, and $\bar{x}^k$ are defined in Algorithm 1.*

According to Algorithm 2, we never need to form or update full-dimensional vectors. Only times that we need $\hat{x}^k$ are when computing the gradient and the dual variable $y_{\beta_{k+1}}^*$. We present two special cases which are common in machine learning, in which we can compute these steps efficiently.

**Remark 3.2.** *Under the following assumptions, we can characterize the per-iteration complexity explicitly. Let $A, M \in \mathbb{R}^{m \times p}$, and*

*(a)  $f$ has the form $f(x) = \sum_{j=1}^m \phi_j(e_j^\top Mx)$, where $e_j$ is the $j^{th}$ standard unit vector.*

*(b)  $h$ is separable as in $h(Ax) = \delta_{\{c\}}(Ax)$ or $h(Ax) = \|Ax\|_1$.*

*Assuming that we store and maintain the residuals $r_{u,f}^k = Mu^k$, $r_{\tilde{z},f}^k = M\tilde{z}^k$, $r_{u,h}^k = Au^k$, $r_{\tilde{z},h}^k = A\tilde{z}^k$, then we have the per-iteration cost as $\mathcal{O}(\max\{|\{j \mid A_{ji} \neq 0\}|, |\{j \mid M_{ji} \neq 0\}|\})$ arithmetic operations. If $h$ is partially separable as in [3], then the complexity of each iteration will remain moderate.*

## 3.3 Case 1: Convergence analysis of SMART-CD for Lipschitz continuous $h$

We provide the following main theorem, which characterizes the convergence rate of Algorithm 1.

**Theorem 3.3.** *Let $x^\star$ be an optimal solution of (1) and let $\beta_1 > 0$ be given. In addition, let $\tau_0 := \min\{q_i \mid i \in [n]\} \in (0, 1]$ and $\beta_0 := (1 + \tau_0)\beta_1$ be given parameters. For all $k \ge 1$, the sequence $\{\bar{x}^k\}$ generated by Algorithm 1 satisfies:*

$$\mathbb{E}\left[F(\bar{x}^k) - F^\star\right] \le \frac{C^*(x^0)}{\tau_0(k-1) + 1} + \frac{\beta_1(1 + \tau_0)D_{h^*}^2}{2(\tau_0 k + 1)}, \tag{8}$$

*where $C^*(x^0) := (1 - \tau_0)(F_{\beta_0}(x^0) - F^\star) + \sum_{i=1}^n \frac{\tau_0 B_i^0}{2q_i}\|x_i^\star - x_i^0\|_{(i)}^2$ and $D_{h^*}$ is as defined by (6).*

In the special case when we use uniform distribution, $\tau_0 = q_i = 1/n$, the convergence rate reduces to

$$\mathbb{E}\left[F(\bar{x}^k) - F^\star\right] \le \frac{nC^*(x^0)}{k+n-1} + \frac{(n+1)\beta_0 D_{h^*}^2}{2k+2n},$$

where $C^*(x^0) := (1 - \frac{1}{n})(F_{\beta_0}(x^0) - F^\star) + \sum_{i=1}^n \frac{B_i^0}{2}\|x_i^\star - x_i^0\|_{(i)}^2$. This estimate shows that the convergence rate of Algorithm 1 is

$$\mathcal{O}\left(\frac{n}{k}\right),$$

which is the best known so far to the best of our knowledge.

### 3.4 Case 2: Convergence analysis of SMART-CD for non-smooth constrained optimization

In this section, we instantiate Algorithm 1 to solve constrained convex optimization problem with possibly non-smooth terms in the objective. Clearly, if we choose $h(\cdot) = \delta_{\{c\}}(\cdot)$ in (1) as the indicator function of the set $\{c\}$ for a given vector $c \in \mathbb{R}^m$, then we obtain a constrained problem:

$$F^\star := \min_{x \in \mathbb{R}^p} \{F(x) = f(x) + g(x) \mid Ax = c\}, \tag{9}$$

where $f$ and $g$ are defined as in (1), $A \in \mathbb{R}^{m \times p}$, and $c \in \mathbb{R}^m$.

We can specify Algorithm 1 to solve this constrained problem by modifying the following two steps:
  (a) The update of $y_{\beta_{k+1}}^*(A\hat{x}^k)$ at Step 5 is changed to

$$y_{\beta_{k+1}}^*(A\hat{x}^k) := \dot{y} + \tfrac{1}{\beta_{k+1}}(A\hat{x}^k - c), \tag{10}$$

   which requires one matrix-vector multiplication in $A\hat{x}^k$.
  (b) The update of $\tau_k$ at Step 9 and $\beta_{k+1}$ at Step 10 are changed to

$$\tau_{k+1} := \tfrac{\tau_k}{1+\tau_k} \quad \text{and} \quad \beta_{k+2} := (1 - \tau_{k+1})\beta_{k+1}. \tag{11}$$

Now, we analyze the convergence of this algorithm by providing the following theorem.

**Theorem 3.4.** *Let $\{\bar{x}^k\}$ be the sequence generated by Algorithm 1 for solving (9) using the updates (10) and (11) and let $y^\star$ be an arbitrary optimal solution of the dual problem of (9). In addition, let $\tau_0 := \min\{q_i \mid i \in [n]\} \in (0,1]$ and $\beta_0 := (1 + \tau_0)\beta_1$ be given parameters. Then, we have the following estimates:*

$$\begin{cases} \mathbb{E}\left[F(\bar{x}^k) - F^\star\right] & \le \frac{C^*(x^0)}{\tau_0(k-1)+1} + \frac{\beta_1\|y^\star - \dot{y}\|^2}{2(\tau_0(k-1)+1)} + \|y^\star\|\mathbb{E}\left[\|A\bar{x}^k - b\|\right], \\[2mm] \mathbb{E}\left[\|A\bar{x}^k - b\|\right] & \le \frac{\beta_1}{\tau_0(k-1)+1}\left[\|y^\star - \dot{y}\| + \left(\|y^\star - \dot{y}\|^2 + 2\beta_1^{-1}C^*(x^0)\right)^{1/2}\right], \end{cases} \tag{12}$$

*where $C^*(x^0) := (1 - \tau_0)(F_{\beta_0}(x^0) - F^\star) + \sum_{i=1}^n \frac{\tau_0 B_i^0}{2q_i}\|x_i^\star - x_i^0\|_{(i)}^2$. We note that the following lower bound always holds $-\|y^\star\|\mathbb{E}\left[\|A\bar{x}^k - b\|\right] \le \mathbb{E}\left[F(\bar{x}^k) - F^\star\right]$.*

### 3.5 Other special cases

We consider the following special cases of Algorithm 1:

**The case $h = 0$:** In this case, we obtain an algorithm similar to the one studied in [7] except that we have non-uniform sampling instead of importance sampling. If the distribution is uniform, then we obtain the method in [4].

**The case $g = 0$:** In this case, we have $F(x) = f(x) + h(Ax)$, which can handle the linearly constrained problems with smooth objective function. In this case, we can choose $\tau_0 = 1$, and the coordinate proximal gradient step, Step 7 in Algorithm 1, is simplified as

$$\tilde{x}_{i_k}^{k+1} := \tilde{x}_{i_k}^k - \frac{q_{i_k}}{\tau_k B_{i_k}^k} H_{i_k}^{-1}\left(\nabla_{i_k} f(\hat{x}^k) + A_{i_k}^\top y_{\beta_{k+1}}^*(\hat{u}^k)\right). \tag{13}$$

In addition, we replace Step 8 with

$$\bar{x}_i^{k+1} = \hat{x}_i^k + \frac{\tau_k}{q_i}(\tilde{x}_i^{k+1} - \tilde{x}_i^k), \quad \forall i \in [n]. \tag{14}$$

We then obtain the following results:

**Corollary 3.5.** *Assume that Assumption 1 holds. Let $\tau_0 = 1$, $\beta_1 > 0$ and Step 7 and 8 of Algorithm 1 be updated by* (13) *and* (14)*, respectively. If, in addition, $h$ is Lipschitz continuous, then we have*

$$\mathbb{E}\left[F(\bar{x}^k) - F^\star\right] \leq \frac{1}{k}\sum_{i=1}^{n}\frac{B_i^0}{2q_i^2}\|x_i^\star - x_i^0\|_{(i)}^2 + \frac{\beta_1 D_{h^*}^2}{k+1}, \tag{15}$$

*where $D_{h^*}$ is defined by* (6).

*If, instead of Lipschitz continuous $h$, we have $h(\cdot) = \delta_{\{c\}}(\cdot)$ to solve the constrained problem* (9) *with $g = 0$, then we have*

$$\begin{cases} \mathbb{E}\left[F(\bar{x}^k) - F^\star\right] & \leq \frac{C^*(x^0)}{k} + \frac{\beta_1\|y^\star - \dot{y}\|^2}{2k} + \|y^\star\|\mathbb{E}\left[\|A\bar{x}^k - b\|\right], \\[2mm] \mathbb{E}\left[\|A\bar{x}^k - b\|\right] & \leq \frac{\beta_1}{k}\left[\|y^\star - \dot{y}\| + \left(\|y^\star - \dot{y}\|^2 + 2\beta_1^{-1}C^*(x^0)\right)^{1/2}\right], \end{cases} \tag{16}$$

*where $C^*(x^0) := \sum_{i=1}^{n}\frac{B_i^0}{2q_i^2}\|x_i^\star - x_i^0\|_{(i)}^2$.*

### 3.6 Restarting SMART-CD

It is known that restarting an accelerated method significantly enhances its practical performance when the underlying problem admits a (restricted) strong convexity condition. As a result, we describe below how to restart (i.e., the momentum term) in Efficient SMART-CD. If the restart is injected in the $k$-th iteration, then we restart the algorithm with the following steps:

$$\begin{cases} u^{k+1} & \leftarrow 0, \\ r_{u,f}^{k+1} & \leftarrow 0, \\ r_{u,h}^{k+1} & \leftarrow 0, \\ \dot{y} & \leftarrow y_{\beta_{k+1}}^*(c_k r_{u,h}^k + r_{\tilde{z},h}^k), \\ \beta_{k+1} & \leftarrow \beta_1, \\ \tau_{k+1} & \leftarrow \tau_0, \\ c_k & \leftarrow 1. \end{cases}$$

The first three steps of the restart procedure is for restarting the primal variable which is classical [15]. Restarting $\dot{y}$ is also suggested in [9]. The cost of this procedure is essentially equal to the cost of one iteration as described in Remark 3.2, therefore even restarting once every epoch will not cause a significant difference in terms of per-iteration cost.

## 4 Numerical evidence

We illustrate the performance of Efficient SMART-CD in brain imaging and support vector machines applications. We also include one representative example of a degenerate linear program to illustrate why the convergence rate guarantees of our algorithm matter. We compare SMART-CD with Vu-Condat-CD [11], which is a coordinate descent variant of Vu-Condat's algorithm [16], FISTA [17], ASGARD [9], Chambolle-Pock's primal-dual algorithm [18], L-BFGS [19] and SDCA [5].

### 4.1 A degenerate linear program: Why do convergence rate guarantees matter?

We consider the following degenerate linear program studied in [9]:

$$\begin{cases} \min_{x \in \mathbb{R}^p} & 2x_p \\ \text{s.t.} & \sum_{k=1}^{p-1}x_k = 1, \\ & x_p - \sum_{k=1}^{p-1}x_k = 0, \qquad (2 \leq j \leq d), \\ & x_p \geq 0. \end{cases} \tag{17}$$

Here, the constraint $x_p - \sum_{k=1}^{p-1}x_k = 0$ is repeated $d$ times. This problem satisfies the linear constraint qualification condition, which guarantees the primal-dual optimality. If we define

$$f(x) = 2x_p, \quad g(x) = \delta_{\{x_p \geq 0\}}(x_p), \quad h(Ax) = \delta_{\{c\}}(Ax),$$

where

$$Ax = \left[\sum_{k=1}^{p-1}x_k, \; x_p - \sum_{k=1}^{p-1}x_k, \ldots, \; x_p - \sum_{k=1}^{p-1}x_k\right]^\top, \quad c = [1, 0, \ldots, 0]^\top,$$

we can fit this problem and its dual form into our template (1).

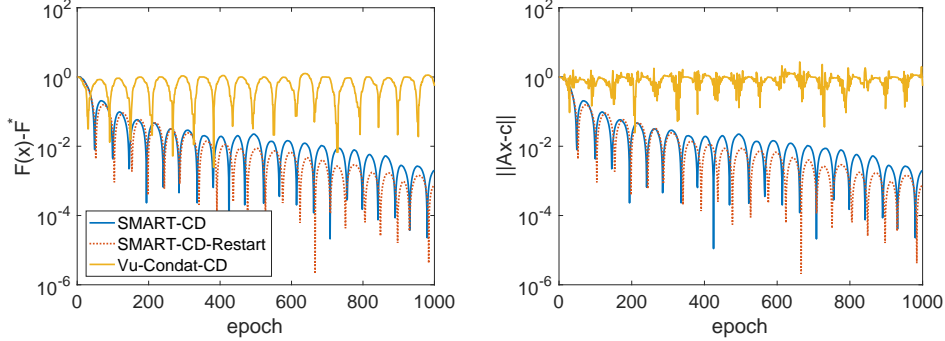
Figure 1: The convergence behavior of 3 algorithms on a degenerate linear program.

For this experiment, we select the dimensions $p = 10$ and $d = 200$. We implement our algorithm and compare it with Vu-Condat-CD. We also combine our method with the restarting strategy proposed above. We use the same mapping to fit the problem into the template of Vu-Condat-CD.

Figure 1 illustrates the convergence behavior of Vu-Condat-CD and SMART-CD. We compare primal suboptimality and feasibility in the plots. The explicit solution of the problem is used to generate the plot with primal suboptimality. We observe that degeneracy of the problem prevents Vu-Condat-CD from making any progress towards the solution, where SMART-CD preserves $\mathcal{O}(1/k)$ rate as predicted by theory. We emphasize that the authors in [11] proved almost sure convergence for Vu-Condat-CD but they did not provide a convergence rate guarantee for this method. Since the problem is certainly non-strongly convex, restarting does not significantly improve performance of SMART-CD.

**4.2 Total Variation and $\ell_1$-regularized least squares regression with functional MRI data**

In this experiment, we consider a computational neuroscience application where prediction is done based on a sequence of functional MRI images. Since the images are high dimensional and the number of samples that can be taken is limited, TV-$\ell_1$ regularization is used to get stable and predictive estimation results [20]. The convex optimization problem we solve is of the form:

$$\min_{x \in \mathbb{R}^p} \tfrac{1}{2}\|Mx - b\|^2 + \lambda r \|x\|_1 + \lambda(1 - r)\|x\|_{\text{TV}}. \tag{18}$$

This problem fits to our template with

$$f(x) = \tfrac{1}{2}\|Mx - b\|^2, \qquad g(x) = \lambda r \|x\|_1, \qquad h(u) = \lambda(1 - r)\|u\|_1,$$

where $D$ is the 3D finite difference operator to define a total variation norm $\|\cdot\|_{\text{TV}}$ and $u = Dx$.

We use an fMRI dataset where the primal variable $x$ is 3D image of the brain that contains 33177 voxels. Feature matrix $M$ has 768 rows, each representing the brain activity for the corresponding example [20]. We compare our algorithm with Vu-Condat's algorithm, FISTA, ASGARD, Chambolle-Pock's primal-dual algorithm, L-BFGS and Vu-Condat-CD.

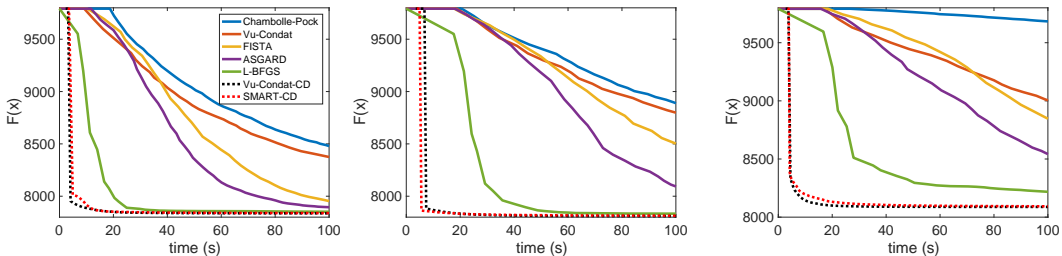
Figure 2: The convergence of 7 algorithms for problem (18). The regularization parameters for the first plot are $\lambda = 0.001, r = 0.5$, for the second plot are $\lambda = 0.001, r = 0.9$, for the third plot are $\lambda = 0.01, r = 0.5$.

Figure 2 illustrates the convergence behaviour of the algorithms for different values of the regularization parameters. Per-iteration cost of SMART-CD and Vu-Condat-CD is similar, therefore the behavior of these two algorithms are quite similar in this experiment. Since Vu-Condat's,

Chambolle-Pock's, FISTA and ASGARD methods work with full dimensional variables, they have slow convergence in time. L-BFGS has a close performance to coordinate descent methods.

The simulation in Figure 2 is performed using benchmarking tool of [20]. The algorithms are tuned for the best parameters in practice.

### 4.3 Linear support vector machines problem with bias

In this section, we consider an application of our algorithm to support vector machines (SVM) problem for binary classification. Given a training set with $m$ examples $\{a_1, a_2, \ldots, a_m\}$ such that $a_i \in \mathbb{R}^p$ and class labels $\{b_1, b_2, \ldots b_m\}$ such that $b_i \in \{-1, +1\}$, we define the soft margin primal support vector machines problem with bias as

$$\min_{w \in \mathbb{R}^p} \sum_{i=1}^{m} C_i \max\left(0, 1 - b_i(\langle a_i, w \rangle + w_0)\right) + \tfrac{\lambda}{2}\|w\|^2. \tag{19}$$

As it is a common practice, we solve its dual formulation, which is a constrained problem:

$$\begin{cases} \min_{x \in \mathbb{R}^m} & \left\{ \tfrac{1}{2\lambda}\|MD(b)x\|^2 - \sum_{i=1}^{m} x_i \right\} \\ \text{s.t.} & 0 \le x_i \le C_i, \ i = 1, \cdots, m, \ b^\top x = 0, \end{cases} \tag{20}$$

where $D(b)$ represents a diagonal matrix that has the class labels $b_i$ in its diagonal and $M \in \mathbb{R}^{p \times m}$ is formed by the example vectors. If we define

$$f(x) = \frac{1}{2\lambda}\|MD(b)x\|^2 - \sum_{i=1}^{m} x_i, \quad g_i(x_i) = \delta_{\{0 \le x_i \le C_i\}}, \quad c = 0, \quad A = b^\top,$$

then, we can fit this problem into our template in (9).

We apply the specific version of SMART-CD for constrained setting from Section 3.4 and compare with Vu-Condat-CD and SDCA. Even though SDCA is a state-of-the-art method for SVMs, we are not able to handle the bias term using SDCA. Hence, it only applies to (20) when $b^\top x = 0$ constraint is removed. This causes SDCA not to converge to the optimal solution when there is bias term in the problem (19). The following table summarizes the properties of the classification datasets we used.

| Data Set | Training Size | Number of Features | Convergence Plot |
|---|---|---|---|
| rcv1.binary [21, 22] | 20,242 | 47,236 | Figure 3, plot 1 |
| a8a [21, 23] | 22,696 | 123 | Figure 3, plot 2 |
| gisette [21, 24] | 6,000 | 5,000 | Figure 3, plot 3 |

Figure 3 illustrates the performance of the algorithms for solving the dual formulation of SVM in (20). We compute the duality gap for each algorithm and present the results with epochs in the horizontal axis since per-iteration complexity of the algorithms is similar. As expected, SDCA gets stuck at a low accuracy since it ignores one of the constraints in the problem. We demonstrate this fact in the first experiment and then limit the comparison to SMART-CD and Vu-Condat-CD. Equipped with restart strategy, SMART-CD shows the fastest convergence behavior due to the restricted strong convexity of (20).

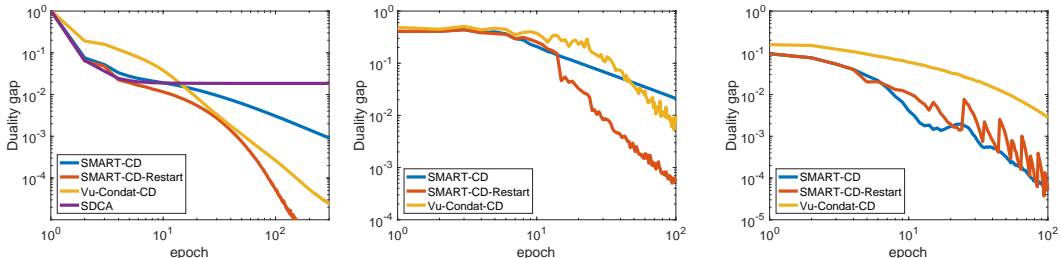

Figure 3: The convergence of $4$ algorithms on the dual SVM (20) with bias. We only used SDCA in the first dataset since it stagnates at a very low accuracy.

## 5 Conclusions

Coordinate descent methods have been increasingly deployed to tackle huge scale machine learning problems in recent years. The most notable works include [1–6]. Our method relates to several works

in the literature including [1, 4, 7, 9, 10, 12]. The algorithms developed in [2–4] only considered a special case of (1) with $h = 0$, and cannot be trivially extended to apply to general setting (1). Here, our algorithm can be viewed as an adaptive variant of the method developed in [4] extended to the sum of three functions. The idea of homotopy strategies relate to [9] for first-order primal-dual methods. This paper further extends such an idea to randomized coordinate descent methods for solving (1). We note that a naive application of the method developed in [4] to the smoothed problem with a carefully chosen fixed smoothness parameter would result in the complexity $\mathcal{O}(n^2/k)$, whereas using our homotopy strategy on the smoothness parameter, we reduced this complexity to $\mathcal{O}(n/k)$.

With additional strong convexity assumption on problem template (1), it is possible to obtain $\mathcal{O}(1/k^2)$ rate by using deterministic first-order primal-dual algorithms [9, 18]. It remains as future work to incorporate strong convexity to coordinate descent methods for solving nonsmooth optimization problems with a faster convergence rate.

## Acknowledgments

The work of O. Fercoq was supported by a public grant as part of the Investissement d'avenir project, reference ANR-11-LABX-0056-LMH, LabEx LMH. The work of Q. Tran-Dinh was partly supported by NSF grant, DMS-1619884, USA. The work of A. Alacaoglu and V. Cevher was supported by European Research Council (ERC) under the European Union's Horizon 2020 research and innovation programme (grant agreement n$^{\mathrm{o}}$ 725594 - time-data).

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
