[Supplementary Material]

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

## A   Key lemmas

The following properties are key to design the algorithm, whose proofs are very similar to the proof of [9, Lemma 10] by using a different norm, and we omit the proof here. The proof of the last property directly follows by using the explicit form of $h_\beta(u)$ in the special case when $h^*(y) = \langle c, y \rangle$.

**Lemma A.1.** *For any $u, \hat{u} \in \mathbb{R}^m$, the function $h_\beta$ defined by* (4) *satisfies the following properties:*

(a) $h_\beta(\cdot)$ is convex and smooth. Its gradient $\nabla h_\beta(u) = y_\beta^*(u)$ is Lipschitz continuous with the Lipschitz constant $L_{h_\beta} = \frac{1}{\beta}$.

(b) $h_\beta(u) + \langle \nabla h_\beta(u), \hat{u} - u \rangle + \frac{\beta}{2} \| y_\beta^*(u) - y_\beta^*(\hat{u}) \|^2 \leq h_\beta(\hat{u})$.

(c) $h(\hat{u}) \geq h_\beta(u) + \langle \nabla h_\beta(u), \hat{u} - u \rangle + \frac{\beta}{2} \| y_\beta^*(u) - \dot{y} \|^2$.

(d) $h_\beta(u) \leq h_{\bar{\beta}}(u) + \left( \frac{\bar{\beta} - \beta}{2} \right) \| y_\beta^*(u) - \dot{y} \|^2$.

(e) *If $h^*(y) = \langle c, y \rangle$, a linear function, then $h_\beta(u) = h_{\bar{\beta}}(u) + \frac{(\bar{\beta} - \beta)\beta}{2\bar{\beta}} \| y_\beta^*(u) - \dot{y} \|^2$.*

**Lemma A.2.** *The parameters $\{\tau_k\}_{k \geq 0}$ and $\{\beta_k\}_{k \geq 1}$ updated by Steps 9 and 10, respectively, satisfy the following bounds:*

$$\frac{1}{k + \tau_0^{-1}} \leq \tau_k \leq \frac{2}{k + \tau_0^{-1} + 1}, \qquad \beta_k \leq \frac{\beta_1(1 + \tau_0)}{\tau_0 k + 1}. \tag{21}$$

*Proof.* We proceed by induction. By Step 9, we have $\tau_{k-1}^2 = \frac{\tau_k^3 + \tau_k^2}{1 - \tau_k}$. For $k = 0$, the bounds trivially hold since $\tau_0 \leq \frac{1}{n}$. By the inductive assumption, we have $\frac{1}{k-1+\tau_0^{-1}} \leq \tau_{k-1} \leq \frac{2}{k+\tau_0^{-1}}$. Assume toward contradiction that $\tau_k < \frac{1}{k+\tau_0^{-1}}$. Then $\frac{1}{(k-1+\tau_0^{-1})^2} \leq \tau_{k-1}^2 = \frac{\tau_k^3 + \tau_k^2}{1-\tau_k} < \frac{k+1+\tau_0^{-1}}{(k+\tau_0^{-1})^2(k-1+\tau_0^{-1})}$, which is a contradiction. Therefore $\tau_k \geq \frac{1}{k+\tau_0^{-1}}$. For the other side of the inequality, assume toward contradiction that $\tau_k > \frac{2}{k+1+\tau_0^{-1}}$. Then $\frac{4(k+3+\tau_0^{-1})}{(k+1+\tau_0^{-1})^2(k-1+\tau_0^{-1})} < \frac{\tau_k^3 + \tau_k^2}{1-\tau_k} = \tau_{k-1}^2 \leq \frac{4}{(k+\tau_0^{-1})^2}$, which is a contradiction. Therefore, $\tau_k \leq \frac{2}{k+1+\tau_0^{-1}}$.

For $\{\beta_k\}$, we note that $\beta_k = \frac{\beta_{k-1}}{1+\tau_{k-1}} = \beta_1 \prod_{i=1}^{k-1} \frac{1}{1+\tau_i} \leq \beta_1 \prod_{i=1}^{k-1} \frac{i+\tau_0^{-1}}{i+1+\tau_0^{-1}} = \frac{\beta_1(1+\tau_0)}{\tau_0 k + 1}$. $\square$

The following lemma is motivated by [4].

**Lemma A.3.** *Consider the iterates $\{\bar{x}^k, \tilde{x}^k\}_{k \geq 0}$ of Algorithm 1. Then, for $k \geq 0$ and $i \in [n]$, we can write $\{\bar{x}_i^k\}$ as a convex combination of $\{\tilde{x}_i^l\}_{l=0}^k$:*

$$\bar{x}_i^k = \sum_{l=0}^k \gamma_i^{k,l} \tilde{x}_i^l, \tag{22}$$

*where $\gamma_i^{k,l} \geq 0$ and $\sum_{l=0}^k \gamma_i^{k,l} = 1$. Moreover, the coefficients $\gamma_i^{k,l}$ can explicitly be computed as*

$$\gamma_i^{k+1,l} = \begin{cases} (1-\tau_k)\gamma_i^{k,l}, & \text{for } l = 0, \cdots, k-1, \\ (1-\tau_k)\gamma_i^{k,k} + \tau_k - \frac{\tau_k}{\tau_0}, & \text{for } l = k, \\ \frac{\tau_k}{\tau_0}, & \text{for } l = k+1. \end{cases} \tag{23}$$

*Proof.* Now, from the definition of $\bar{x}^{k+1}$ and $\hat{x}^k$, for $i \in [n]$, we can write

$$\bar{x}_i^{k+1} = (1-\tau_k)\bar{x}_i^k + \tau_k \tilde{x}_i^k + \frac{\tau_k}{\tau_0}(\tilde{x}_i^{k+1} - \tilde{x}_i^k) = (1-\tau_k)\bar{x}_i^k + (\tau_k - \frac{\tau_k}{\tau_0})\tilde{x}_i^k + \frac{\tau_k}{\tau_0}\tilde{x}_i^{k+1}. \tag{24}$$

We prove that $\bar{x}_i^k = \sum_{l=0}^k \gamma_i^{k,l}\tilde{x}_i^l$ for $i \in [n]$ such that $\gamma_i^{k,l} \geq 0$ and $\sum_{l=0}^k \gamma_i^{k,l} = 1$. Indeed, for $k = 0$, we have $\bar{x}^0 = \tilde{x}^0$, which trivially holds if we choose $\gamma_i^{0,0} = 1$. Now, assume that this expression holds for $k \geq 1$, we prove it holds for $k + 1$. Indeed, from (24), using this induction assumption, we can write

$$\bar{x}_i^{k+1} = (1 - \tau_k)\sum_{l=0}^{k-1}\gamma_i^{k,l}\tilde{x}_i^l + \left[(1 - \tau_k)\gamma_i^{k,k} + \tau_k - \frac{\tau_k}{\tau_0}\right]\tilde{x}_i^k + \frac{\tau_k}{\tau_0}\tilde{x}_i^{k+1} = \sum_{l=0}^{k+1}\gamma_i^{k+1,l}\tilde{x}_i^l,$$

where constants $\gamma_i^{k+1,l}$ are as given in (23). It is trivial to check that $\sum_{l=0}^{k+1}\gamma_i^{k+1,l} = (1 - \tau_k)\sum_{l=0}^k\gamma_i^{k,l} + \tau_k - \frac{\tau_k}{\tau_0} + \frac{\tau_k}{\tau_0} = (1 - \tau_k) + \tau_k = 1$. In addition, since $\{\tau_k\}_{k\geq 0}$ is a non-increasing sequence, $\gamma_i^{k,l} \geq 0$. $\qquad\square$

# B  Convergence analysis of SMART-CD

## B.1  The proof of Theorem 3.3

First, let us define the full primal proximal-gradient step as

$$\bar{\bar{x}}^{k+1} := \arg\min_{x\in\mathbb{R}^p}\left\{\langle\nabla\psi_{\beta_{k+1}}(\hat{x}^k), x - \hat{x}^k\rangle + g(x) + \tau_k\sum_{i=1}^n\frac{B_i^k}{2\tau_0}\|x_i - \tilde{x}_i^k\|_{(i)}^2\right\}, \qquad (25)$$

where $\nabla\psi_{\beta_{k+1}}(\hat{x}^k) = \nabla f(\hat{x}^k) + A^\top y_{\beta_{k+1}}^*(A\hat{x}^k)$. The primal coordinate step (Step 7) and Step 8 in Algorithm 1 can be written as

$$\tilde{x}_i^{k+1} = \begin{cases}\bar{\bar{x}}_i^{k+1}, & \text{if } i = i_k, \\ \tilde{x}_i^k, & \text{otherwise.}\end{cases} \qquad (26)$$

Moreover, using [25, Property 2], we know that for all $x \in \mathbb{R}^p$ and for all $i \in [n]$,

$$g_i(\bar{\bar{x}}_i^{k+1}) \leq g_i(x_i) + \langle\nabla_i\psi_{\beta_{k+1}}(\hat{x}^k), x_i - \bar{\bar{x}}_i^{k+1}\rangle + \frac{\tau_k B_i^k}{2\tau_0}\left(\|x_i - \tilde{x}_i^k\|_{(i)}^2 - \|x_i - \bar{\bar{x}}_i^{k+1}\|_{(i)}^2\right)$$

$$- \frac{\tau_k B_i^k}{2\tau_0}\|\bar{\bar{x}}_i^{k+1} - \tilde{x}_i^k\|_{(i)}^2. \qquad (27)$$

Now, since the partial gradient $\nabla_{i_k}f$ is $\hat{L}_{i_k}$-Lipschitz continuous, using $\bar{x}_{i_k}^{k+1} = \hat{x}_{i_k}^k + \frac{\tau_k}{\tau_0}(\tilde{x}_{i_k}^{k+1} - \tilde{x}_{i_k}^k)$ and $\bar{x}_i^{k+1} = \hat{x}_i^k$ for $i \neq i_k$, we have

$$f(\bar{x}^{k+1}) \leq f(\hat{x}^k) + \langle\nabla_{i_k}f(\hat{x}^k), \bar{x}_{i_k}^{k+1} - \hat{x}_{i_k}^k\rangle + \frac{\hat{L}_{i_k}}{2}\|\bar{x}_{i_k}^{k+1} - \hat{x}_{i_k}^k\|_{(i_k)}^2$$

$$= f(\hat{x}^k) + \frac{\tau_k}{\tau_0}\langle\nabla_{i_k}f(\hat{x}^k), \tilde{x}_{i_k}^{k+1} - \tilde{x}_{i_k}^k\rangle + \frac{\tau_k^2\hat{L}_{i_k}}{2\tau_0^2}\|\tilde{x}_{i_k}^{k+1} - \tilde{x}_{i_k}^k\|_{(i_k)}^2. \qquad (28)$$

Taking the $\mathcal{F}_k$-conditional expectation with respect to $i_k$ and noting (26), we obtain

$$\mathbb{E}_{i_k}\left[f(\bar{x}^{k+1}) \mid \mathcal{F}_k\right] \leq f(\hat{x}^k) + \frac{\tau_k}{\tau_0}\sum_{i=1}^n q_i\langle\nabla_i f(\hat{x}^k), \bar{\bar{x}}_i^{k+1} - \tilde{x}_i^k\rangle$$

$$+ \frac{\tau_k^2}{\tau_0^2}\sum_{i=1}^n q_i\frac{\hat{L}_i}{2}\|\bar{\bar{x}}_i^{k+1} - \tilde{x}_i^k\|_{(i)}^2. \qquad (29)$$

Next, let us denote by $\varphi_\beta(x) := h_\beta(Ax)$. Then, by Lemma A.1, we can see that $\varphi_{\beta_{k+1}}$ has block-coordinate Lipschitz gradient with the Lipschitz constant $\frac{\|A_i\|^2}{\beta_{k+1}}$, where $A_i$ is the $i$-th column block of $A$. Moreover, $\nabla_i\varphi_{\beta_{k+1}}(x) = A_i^\top y_{\beta_{k+1}}^*(Ax)$. Hence, using $\bar{x}_{i_k}^{k+1} = \hat{x}_{i_k}^k + \frac{\tau_k}{\tau_0}(\tilde{x}_{i_k}^{k+1} - \tilde{x}_{i_k}^k)$ and $\bar{x}_i^{k+1} = \hat{x}_i^k$ for $i \neq i_k$, we can write

$$\varphi_{\beta_{k+1}}(\bar{x}^{k+1}) \leq \varphi_{\beta_{k+1}}(\hat{x}^k) + \langle\nabla_{i_k}\varphi_{\beta_{k+1}}(\hat{x}^k), \bar{x}_{i_k}^{k+1} - \hat{x}_{i_k}^k\rangle + \frac{\|A_i\|^2}{2\beta_{k+1}}\|\bar{x}_{i_k}^{k+1} - \hat{x}_{i_k}^k\|_{(i_k)}^2$$

$$= \varphi_{\beta_{k+1}}(\hat{x}^k) + \frac{\tau_k}{\tau_0}\langle\nabla_{i_k}\varphi_{\beta_{k+1}}(\hat{x}^k), \tilde{x}_{i_k}^{k+1} - \tilde{x}_{i_k}^k\rangle + \frac{\tau_k^2\|A_i\|^2}{2\tau_0^2\beta_{k+1}}\|\tilde{x}_{i_k}^{k+1} - \tilde{x}_{i_k}^k\|_{(i_k)}^2.$$

Taking the $\mathcal{F}_k$-conditional expectation with respect to $i_k$ given $\mathcal{F}_k$ and noting (26), we get

$$\mathbb{E}_{i_k}\left[\varphi_{\beta_{k+1}}(\bar{x}^{k+1}) \mid \mathcal{F}_k\right] \leq \varphi_{\beta_{k+1}}(\hat{x}^k) + \frac{\tau_k}{\tau_0}\sum_{i=1}^n q_i\langle\nabla_i\varphi_{\beta_{k+1}}(\hat{x}^k), \bar{\tilde{x}}_i^{k+1} - \tilde{x}_i^k\rangle$$

$$+ \frac{\tau_k^2}{\tau_0^2}\sum_{i=1}^n q_i\frac{\|A_i\|^2}{2\beta_{k+1}}\|\bar{\tilde{x}}_i^{k+1} - \tilde{x}_i^k\|_{(i)}^2. \tag{30}$$

Now, we define

$$\hat{g}_i^k := \sum_{l=0}^k \gamma_i^{k,l}g_i(\tilde{x}_i^l) \quad\text{and}\quad \hat{g}^k := \sum_{i=1}^n \hat{g}_i^k. \tag{31}$$

Using Lemma A.3, we can write

$$\hat{g}_i^{k+1} = \sum_{l=0}^{k+1}\gamma_i^{k+1,l}g_i(\tilde{x}_i^l)$$

$$= \sum_{l=0}^{k-1}(1-\tau_k)\gamma_i^{k,l}g_i(\tilde{x}_i^l) + \left[(1-\tau_k)\gamma_i^{k,k} + \tau_k - \frac{\tau_k}{\tau_0}\right]g_i(\tilde{x}_i^k) + \frac{\tau_k}{\tau_0}g_i(\tilde{x}_i^{k+1})$$

$$= (1-\tau_k)\sum_{l=0}^k\gamma_i^{k,l}g_i(\tilde{x}_i^l) + \tau_kg_i(\tilde{x}_i^k) + \frac{\tau_k}{\tau_0}\left(g_i(\tilde{x}_i^{k+1}) - g_i(\tilde{x}_i^k)\right)$$

$$= (1-\tau_k)\hat{g}_i^k + \tau_kg_i(\tilde{x}_i^k) + \frac{\tau_k}{\tau_0}\left(g_i(\tilde{x}_i^{k+1}) - g_i(\tilde{x}_i^k)\right).$$

Using the definition (31) of $\hat{g}^k$, this estimate implies

$$\hat{g}^{k+1} = (1-\tau_k)\hat{g}^k + \sum_{i=1}^n\left[\tau_kg_i(\tilde{x}_i^k) + \frac{\tau_k}{\tau_0}\left(g_i(\tilde{x}_i^{k+1}) - g_i(\tilde{x}_i^k)\right)\right].$$

Now, by the expression (26), we can show that

$$\mathbb{E}_{i_k}\left[g_i(\tilde{x}_i^{k+1}) \mid \mathcal{F}_k\right] = q_ig_i(\bar{\tilde{x}}_i^{k+1}) + (1-q_i)g_i(\tilde{x}_i^k).$$

Combining the two last expressions, we can derive

$$\mathbb{E}_{i_k}\left[\hat{g}^{k+1} \mid \mathcal{F}_k\right] = (1-\tau_k)\hat{g}^k + \sum_{i=1}^n\left[\tau_kg_i(\tilde{x}_i^k) + \frac{\tau_k}{\tau_0}\left(\mathbb{E}_{i_k}\left[g_i(\tilde{x}_i^{k+1}) \mid \mathcal{F}_k\right] - g_i(\tilde{x}_i^k)\right)\right]$$

$$= (1-\tau_k)\hat{g}^k + \tau_k\sum_{i=1}^n g_i(\tilde{x}_i^k) + \frac{\tau_k}{\tau_0}\sum_{i=1}^n q_i\left(g_i(\bar{\tilde{x}}_i^{k+1}) - g_i(\tilde{x}_i^k)\right). \tag{32}$$

Let us define $\hat{F}_{\beta_k}^k := f(\bar{x}^k) + \hat{g}^k + h_{\beta_k}(A\bar{x}^k) \equiv f(\bar{x}^k) + \hat{g}^k + \varphi_{\beta_k}(\bar{x}^k)$. Then, from (29), (30) and (32), we have that

$$\mathbb{E}_{i_k}\left[\hat{F}_{\beta_{k+1}}^{k+1} \mid \mathcal{F}_k\right] = \mathbb{E}_{i_k}\left[f(\bar{x}^{k+1}) \mid \mathcal{F}_k\right] + \mathbb{E}_{i_k}\left[\hat{g}^{k+1} \mid \mathcal{F}_k\right] + \mathbb{E}_{i_k}\left[\varphi_{\beta_{k+1}}(\bar{x}^{k+1}) \mid \mathcal{F}_k\right]$$

$$\leq \left[f(\hat{x}^k) + \frac{\tau_k}{\tau_0}\sum_{i=1}^n q_i\langle\nabla_if(\hat{x}^k), \bar{\tilde{x}}_i^{k+1} - \tilde{x}_i^k\rangle\right]$$

$$+ \left[\varphi_{\beta_{k+1}}(\hat{x}^k) + \frac{\tau_k}{\tau_0}\sum_{i=1}^n q_i\langle\nabla_i\varphi_{\beta_{k+1}}(\hat{x}^k), \bar{\tilde{x}}_i^{k+1} - \tilde{x}_i^k\rangle\right]$$

$$+ \left[(1-\tau_k)\hat{g}^k + \tau_k\sum_{i=1}^n g_i(\tilde{x}_i^k) + \frac{\tau_k}{\tau_0}\sum_{i=1}^n q_i\left(g_i(\bar{\tilde{x}}_i^{k+1}) - g_i(\tilde{x}_i^k)\right)\right]$$

$$+ \frac{\tau_k^2}{2\tau_0^2}\sum_{i=1}^n q_i\left(\hat{L}_i + \frac{\|A_i\|^2}{\beta_{k+1}}\right)\|\bar{\tilde{x}}_i^{k+1} - \tilde{x}_i^k\|_{(i)}^2, \tag{33}$$

since $\nabla \psi_{\beta_{k+1}}(\hat{x}^k) = \nabla f(\hat{x}^k) + \nabla \varphi_{\beta_{k+1}}(\hat{x}^k)$. Now, using the estimate (27) into the last expression and noting that $B_i^k = \hat{L}_i + \frac{\|A_i\|^2}{\beta_{k+1}}$, we can further derive that for all $x$,

$$
\begin{aligned}
\mathbb{E}_{i_k}\left[\hat{F}_{\beta_{k+1}}^{k+1} \mid \mathcal{F}_k\right] \leq &\left[f(\hat{x}^k) + \frac{\tau_k}{\tau_0}\sum_{i=1}^n q_i \langle \nabla_i f(\hat{x}^k), x_i - \tilde{x}_i^k \rangle\right] \\
&+ \left[\varphi_{\beta_{k+1}}(\hat{x}^k) + \frac{\tau_k}{\tau_0}\sum_{i=1}^n q_i \langle \nabla_i \varphi_{\beta_{k+1}}(\hat{x}^k), x_i - \tilde{x}_i^k \rangle\right] \\
&+ \left[(1-\tau_k)\hat{g}^k + \tau_k \sum_{i=1}^n g_i(\tilde{x}_i^k) + \frac{\tau_k}{\tau_0}\sum_{i=1}^n q_i \left(g_i(x_i) - g_i(\tilde{x}_i^k)\right)\right] \\
&+ \sum_{i=1}^n q_i \frac{\tau_k^2 B_i^k}{2\tau_0^2}\left(\|x_i - \tilde{x}_i^k\|_{(i)}^2 - \|x_i - \bar{\tilde{x}}_i^{k+1}\|_{(i)}^2\right).
\end{aligned} \tag{34}
$$

Let us choose $x$ such that for all $i \in [n]$, $x_i = \left(1 - \frac{\tau_0}{q_i}\right)\tilde{x}_i^k + \frac{\tau_0}{q_i}x_i^\star$. Note that as $\tau_0 \leq q_i$ for all $i$, $x_i$ is a convex combination of $\tilde{x}_i^k$ and $x_i^\star$. We obtain

$$
\begin{aligned}
\mathbb{E}_{i_k}\left[\hat{F}_{\beta_{k+1}}^{k+1} \mid \mathcal{F}_k\right] \leq &\left[f(\hat{x}^k) + \tau_k \langle \nabla f(\hat{x}^k), x^\star - \tilde{x}^k \rangle\right] + \left[\varphi_{\beta_{k+1}}(\hat{x}^k) + \tau_k \langle \nabla \varphi_{\beta_{k+1}}(\hat{x}^k), x^\star - \tilde{x}^k \rangle\right] \\
&+ \left[(1-\tau_k)\hat{g}^k + \tau_k g(x^\star)\right] \\
&+ \sum_{i=1}^n q_i \frac{\tau_k^2 B_i^k}{2\tau_0^2}\left(\left\|\frac{\tau_0}{q_i}(x_i^\star - \tilde{x}_i^k)\right\|_{(i)}^2 - \left\|\left(1 - \frac{\tau_0}{q_i}\right)\tilde{x}_i^k + \frac{\tau_0}{q_i}x_i^\star - \bar{\tilde{x}}_i^{k+1}\right\|_{(i)}^2\right).
\end{aligned} \tag{35}
$$

We simplify the norm difference using the fact that $\|ax + (1-a)y - z\|^2 = a\|x - z\|^2 + (1-a)\|y - z\|^2 - a(1-a)\|x - y\|^2$.

$$
\begin{aligned}
&\left\|\left(1 - \frac{\tau_0}{q_i}\right)\tilde{x}_i^k + \frac{\tau_0}{q_i}x_i^\star - \bar{\tilde{x}}_i^{k+1}\right\|_{(i)}^2 \\
&= \left(1 - \frac{\tau_0}{q_i}\right)\|\tilde{x}_i^k - \bar{\tilde{x}}_i^{k+1}\|_{(i)}^2 + \frac{\tau_0}{q_i}\|x_i^\star - \bar{\tilde{x}}_i^{k+1}\|_{(i)}^2 - \left(1 - \frac{\tau_0}{q_i}\right)\frac{\tau_0}{q_i}\|\tilde{x}_i^k - x_i^\star\|_{(i)}^2 \\
&\geq \frac{\tau_0}{q_i}\|x_i^\star - \bar{\tilde{x}}_i^{k+1}\|_{(i)}^2 - \left(1 - \frac{\tau_0}{q_i}\right)\frac{\tau_0}{q_i}\|\tilde{x}_i^k - x_i^\star\|_{(i)}^2.
\end{aligned}
$$

and we get

$$
\begin{aligned}
\mathbb{E}_{i_k}\left[\hat{F}_{\beta_{k+1}}^{k+1} \mid \mathcal{F}_k\right] \leq &\left[f(\hat{x}^k) + \tau_k \langle \nabla f(\hat{x}^k), x^\star - \tilde{x}^k \rangle\right] + \left[\varphi_{\beta_{k+1}}(\hat{x}^k) + \tau_k \langle \nabla \varphi_{\beta_{k+1}}(\hat{x}^k), x^\star - \tilde{x}^k \rangle\right] \\
&+ \left[(1-\tau_k)\hat{g}^k + \tau_k g(x^\star)\right] + \sum_{i=1}^n \frac{\tau_k^2 B_i^k}{2\tau_0}\left(\|x_i^\star - \tilde{x}_i^k\|_{(i)}^2 - \|\bar{\tilde{x}}_i^{k+1} - x_i^\star\|_{(i)}^2\right).
\end{aligned} \tag{36}
$$

Using the convexity of $f$, we have $f(\hat{x}^k) + \langle \nabla f(\hat{x}^k), x^\star - \hat{x}^k \rangle \leq f(x^\star)$ and $f(\hat{x}^k) + \langle \nabla f(\hat{x}^k), \bar{x}^k - \hat{x}^k \rangle \leq f(\bar{x}^k)$. Moreover, since $\hat{x}^k = (1 - \tau_k)\bar{x}^k + \tau_k \tilde{x}^k$, we have $\tau_k(x^\star - \tilde{x}^k) = (1 - \tau_k)(\bar{x}^k - \hat{x}^k) + \tau_k(x^\star - \hat{x}^k)$. Combining these expressions, we obtain

$$
f(\hat{x}^k) + \tau_k \langle \nabla f(\hat{x}^k), x^\star - \tilde{x}^k \rangle \leq (1 - \tau_k)f(\bar{x}^k) + \tau_k f(x^\star). \tag{37}
$$

On the one hand, by the Lipschitz gradient and convexity of $\varphi_{\beta_{k+1}}$ in Lemma A.1(b), we have

$$
\varphi_{\beta_{k+1}}(\hat{x}^k) + \langle \nabla \varphi_{\beta_{k+1}}(\hat{x}^k), \bar{x}^k - \hat{x}^k \rangle \leq \varphi_{\beta_{k+1}}(\bar{x}^k) - \frac{\beta_{k+1}}{2}\|y_{\beta_{k+1}}^*(A\hat{x}^k) - y_{\beta_{k+1}}^*(A\bar{x}^k)\|^2.
$$

On the other hand, by Lemma A.1(c), we also have

$$
\varphi_{\beta_{k+1}}(\hat{x}^k) + \langle \nabla \varphi_{\beta_{k+1}}(\hat{x}^k), x^\star - \hat{x}^k \rangle \leq h(Ax^\star) - \frac{\beta_{k+1}}{2}\|y_{\beta_{k+1}}^*(A\hat{x}^k) - \dot{y}\|^2
$$

Combining these two inequalities and using $\tau_k(x^\star - \tilde{x}^k) = (1 - \tau_k)(\bar{x}^k - \hat{x}^k) + \tau_k(x^\star - \hat{x}^k)$, we get

$$\varphi_{\beta_{k+1}}(\hat{x}^k) + \tau_k\langle\nabla\varphi_{\beta_{k+1}}(\hat{x}^k), x^\star - \tilde{x}^k\rangle \leq (1 - \tau_k)\varphi_{\beta_{k+1}}(\bar{x}^k) + \tau_k h(Ax^\star)$$
$$- \frac{(1 - \tau_k)\beta_{k+1}}{2}\|y^*_{\beta_{k+1}}(A\hat{x}^k) - y^*_{\beta_{k+1}}(A\bar{x}^k)\|^2 - \frac{\tau_k\beta_{k+1}}{2}\|y^*_{\beta_{k+1}}(A\hat{x}^k) - \dot{y}\|^2.$$

Next, using Lemma A.1(d), we can further estimate

$$\varphi_{\beta_{k+1}}(\hat{x}^k) + \tau_k\langle\nabla\varphi_{\beta_{k+1}}(\hat{x}^k), x^\star - \tilde{x}^k\rangle \leq (1 - \tau_k)\varphi_{\beta_k}(\bar{x}^k) + \tau_k h(Ax^\star)$$
$$- \frac{(1 - \tau_k)\beta_{k+1}}{2}\|y^*_{\beta_{k+1}}(A\hat{x}^k) - y^*_{\beta_{k+1}}(A\bar{x}^k)\|^2 - \frac{\tau_k\beta_{k+1}}{2}\|y^*_{\beta_{k+1}}(A\hat{x}^k) - \dot{y}\|^2$$
$$+ \frac{(1 - \tau_k)(\beta_k - \beta_{k+1})}{2}\|y^*_{\beta_{k+1}}(A\bar{x}^k) - \dot{y}\|^2$$
$$\leq (1 - \tau_k)\varphi_{\beta_k}(\bar{x}^k) + \tau_k h(Ax^\star)$$
$$- \frac{1}{2}\left(\beta_{k+1}\tau_k(1 - \tau_k) - (1 - \tau_k)(\beta_k - \beta_{k+1})\right)\|y^*_{\beta_{k+1}}(A\bar{x}^k) - \dot{y}\|^2. \tag{38}$$

Here, in the last inequality, we use the fact that $(1 - \tau)\|a - b\|^2 + \tau\|a\|^2 - \tau(1 - \tau)\|b\|^2 = \|a - (1 - \tau)b\|^2 \geq 0$ for any $a$, $b$, and $\tau \in [0, 1]$. Substituting (37) and (38) into (36), we obtain

$$\mathbb{E}_{i_k}\left[\hat{F}^{k+1}_{\beta_{k+1}} \mid \mathcal{F}_k\right] \leq (1 - \tau_k)\left[f(\bar{x}^k) + \hat{g}^k + \varphi_{\beta_k}(\bar{x}^k)\right] + \tau_k\left[f(x^\star) + g(x^\star) + h(Ax^\star)\right]$$
$$+ \sum_{i=1}^n \frac{\tau_k^2 B_i^k}{2\tau_0}\left(\|x_i^\star - \tilde{x}_i^k\|_{(i)}^2 - \|x_i^\star - \bar{\tilde{x}}_i^{k+1}\|_{(i)}^2\right)$$
$$- \frac{(1 - \tau_k)}{2}\left[\beta_{k+1}(1 + \tau_k) - \beta_k\right]\|y^*_{\beta_{k+1}}(A\bar{x}^k) - \dot{y}\|^2. \tag{39}$$

Next, let us denote by $Q_k := \sum_{i=1}^n \frac{\tau_k^2 B_i^k}{2\tau_0}\left[\|x_i^\star - \tilde{x}_i^k\|_{(i)}^2 - \|x_i^\star - \bar{\tilde{x}}_i^{k+1}\|_{(i)}^2\right]$. We can further express $Q_k$ as

$$Q_k = \sum_{i=1}^n \frac{\tau_k^2 B_i^k}{2\tau_0}\left[\|x_i^\star - \tilde{x}_i^k\|_{(i)}^2 - \|x_i^\star - \bar{\tilde{x}}_i^{k+1}\|_{(i)}^2\right]$$
$$= \mathbb{E}_{i_k}\left[\frac{\tau_k^2 B_{i_k}^k}{2q_{i_k}\tau_0}\left(\|x_{i_k}^\star - \tilde{x}_{i_k}^k\|_{(i_k)}^2 - \|x_{i_k}^\star - \tilde{x}_{i_k}^{k+1}\|_{(i_k)}^2\right) \mid \mathcal{F}_k\right]$$
$$= \mathbb{E}_{i_k}\left[\sum_{i=1}^n \frac{\tau_k^2 B_i^k}{2q_i\tau_0}\left(\|x_i^\star - \tilde{x}_i^k\|_{(i)}^2 - \|x_i^\star - \tilde{x}_i^{k+1}\|_{(i)}^2\right) \mid \mathcal{F}_k\right], \tag{40}$$

where the last equality follows from the fact that $\tilde{x}_i^{k+1} = \tilde{x}_i^k$ for $i \neq i_k$.

Substituting this expression into (39) and using the definition of $\hat{F}^k_{\beta_k}$ and $F^\star := F(x^\star) = f(x^\star) + g(x^\star) + h(Ax^\star)$, we get

$$\mathbb{E}_{i_k}\left[\hat{F}^{k+1}_{\beta_{k+1}} + \sum_{i=1}^n \frac{\tau_k^2 B_i^k}{2q_i\tau_0}\|x_i^\star - \tilde{x}_i^{k+1}\|_{(i)}^2 \mid \mathcal{F}_k\right] \leq (1 - \tau_k)\hat{F}^k_{\beta_k} + \tau_k F(x^\star)$$
$$+ \sum_{i=1}^n \frac{\tau_k^2 B_i^k}{2q_i\tau_0}\|x_i^\star - \tilde{x}_i^k\|_{(i)}^2 - \mathcal{R}_k,$$

where $\mathcal{R}_k := \frac{(1 - \tau_k)}{2}\left[\beta_{k+1}(1 + \tau_k) - \beta_k\right]\|y^*_{\beta_{k+1}}(A\bar{x}^k) - \dot{y}\|^2$. Assume that we choose $\beta_k$ and $\tau_k$ such that $\beta_{k+1}(1 + \tau_k) - \beta_k \geq 0$, then $\mathcal{R}_k \geq 0$. Taking the expected value of the last estimate over the $\sigma$-field $\mathcal{F}_k$, we obtain

$$\mathbb{E}\left[\hat{F}^{k+1}_{\beta_{k+1}} - F^\star\right] + \mathbb{E}\left[\sum_{i=1}^n \frac{\tau_k^2 B_i^k}{2q_i\tau_0}\|x_i^\star - \tilde{x}_i^{k+1}\|_{(i)}^2\right] \leq (1 - \tau_k)\mathbb{E}\left[\hat{F}^k_{\beta_k} - F^\star\right]$$
$$+ \mathbb{E}\left[\sum_{i=1}^n \frac{\tau_k^2 B_i^k}{2q_i\tau_0}\|x_i^\star - \tilde{x}_i^k\|_{(i)}^2\right]. \tag{41}$$

In order to telescope this inequality we assume that $\tau_k^2 B_i^k \leq (1 - \tau_k)\left(\tau_{k-1}^2 B_i^{k-1}\right)$, which is equivalent to

$$\tau_k^2\left(\hat{L}_i + \frac{\|A_i\|^2}{\beta_{k+1}}\right) \leq (1 - \tau_k)\left[\tau_{k-1}^2\left(\hat{L}_i + \frac{\|A_i\|^2}{\beta_k}\right)\right]. \tag{42}$$

Let us update $\beta_{k+1} = \frac{\beta_k}{1+\tau_k}$. Then, this condition becomes

$$\tau_k^2\left(\beta_k\hat{L}_i + (1+\tau_k)\|A_i\|^2\right) \leq (1-\tau_k)\tau_{k-1}^2\left(\beta_k\hat{L}_i + \|A_i\|^2\right). \tag{43}$$

The condition (43) holds if $\tau_k^2(1 + \tau_k) = (1 - \tau_k)\tau_{k-1}^2$. Hence, we can compute $\tau_k$ as the unique positive root of $\tau^3 + \tau^2 + \tau_{k-1}^2\tau - \tau_{k-1}^2 = 0$. By Lemma A.2, the root of this cubic satisfies $\frac{1}{k+\tau_0^{-1}} \leq \tau_k \leq \frac{2}{k+\tau_0^{-1}+1}$. Let us define $S_k = \sum_{i=1}^n \frac{\tau_k^2 B_i^k}{2q_i\tau_0}\|x_i^\star - \tilde{x}_i^{k+1}\|_{(i)}^2$. Then, we can recursively show that

$$\mathbb{E}\left[\hat{F}_{\beta_{k+1}}^{k+1} - F^\star + S_k\right] \leq \prod_{i=1}^k(1 - \tau_i)\mathbb{E}\left[\hat{F}_{\beta_1}^1 - F^\star + \sum_{i=1}^n\frac{\tau_0^2 B_i^0}{2q_i\tau_0}\|x_i^\star - \tilde{x}_i^1\|_{(i)}^2\right]$$

$$\leq \prod_{i=1}^k(1 - \tau_i)\left((1 - \tau_0)(\hat{F}_{\beta_0}^0 - F^\star) + \sum_{i=1}^n\frac{\tau_0^2 B_i^0}{2q_i\tau_0}\|x_i^\star - \tilde{x}_i^0\|_{(i)}^2\right),$$

where the second inequality follows from (41). Since $\tau_k \geq \frac{1}{k+\tau_0^{-1}}$, it is trivial to show that $\omega_{k+1} := \prod_{i=1}^k(1 - \tau_i) \leq \prod_{i=1}^k\frac{i+\tau_0^{-1}-1}{i+\tau_0^{-1}} = \frac{1}{\tau_0 k+1}$. Now, we have $F_{\beta_0}(x^0) = f(x^0) + g(x^0) + h_{\beta_0}(Ax^0) = \hat{F}_{\beta_0}^0$, and $\tilde{x}^0 = x^0$. In addition, by the convexity of $g$ and Lemma A.3, we also have $g(\bar{x}^k) = g\left(\sum_{l=0}^k\gamma^{k,l}\tilde{x}^l\right) \leq \sum_{l=0}^k\gamma^{k,l}g(\tilde{x}^l) = \hat{g}^k$. Hence, we can write the above estimate as

$$\mathbb{E}\left[F_{\beta_k}(\bar{x}^k) - F^\star\right] \leq \frac{1}{\tau_0(k - 1) + 1}\left[(1 - \tau_0)(F_{\beta_0}(x^0) - F^\star) + \sum_{i=1}^n\frac{\tau_0 B_i^0}{2q_i}\|x_i^\star - x_i^0\|_{(i)}^2\right]. \tag{44}$$

Now, using the bound (6), we have $0 \leq F(\bar{x}^k) - F^\star \leq F_{\beta_k}(\bar{x}^k) - F^\star + \beta_k\frac{D_{h^*}^2}{2}$. Combining this estimate and the above inequality, and noting that $\beta_k \leq \frac{\beta_1(1+\tau_0)}{\tau_0 k+1}$ by Lemma A.2, we obtain the bound in (8). $\qquad\square$

## B.2 The proof of Theorem 3.4

Since $h(u) = \delta_{\{c\}}(u)$, we can smooth this function as $h_\beta(u) = \max_y\left\{\langle u - c, y\rangle - \frac{\beta}{2}\|y - \dot{y}\|^2\right\}$. Let us first define $S_\beta(x) := \mathbb{E}\left[F(x) + h_\beta(Ax) - F^\star\right]$. Since $h^*(y) = \langle c, y\rangle$, we use Lemma A.1(e) to estimate (38) in the proof of Theorem 3.3 instead of Lemma A.1(d) to obtain

$$\varphi_{\beta_{k+1}}(\hat{x}^k) + \tau_k\langle\nabla\varphi_{\beta_{k+1}}(\hat{x}^k), x^\star - \tilde{x}^k\rangle \leq (1 - \tau_k)\varphi_{\beta_k}(\bar{x}^k) + \tau_k h(Ax^\star)$$

$$- \frac{(1 - \tau_k)\beta_{k+1}}{2\beta_k}\left[\beta_{k+1} - (1 - \tau_k)\beta_k\right]\|y_{\beta_{k+1}}^*(A\bar{x}^k) - \dot{y}\|^2. \tag{45}$$

Hence, if $\beta_{k+1} = (1 - \tau_k)\beta_k$, then $\varphi_{\beta_{k+1}}(\hat{x}^k) + \tau_k\langle\nabla\varphi_{\beta_{k+1}}(\hat{x}^k), x^\star - \tilde{x}^k\rangle \leq (1 - \tau_k)\varphi_{\beta_k}(\bar{x}^k) + \tau_k h(Ax^\star)$. Now, we combine the condition $\beta_{k+1} = (1 - \tau_k)\beta_k$ and (42), we can show that

$$\tau_k^2\left((1 - \tau_k)\beta_k\hat{L}_i + \|A_i\|^2\right) \leq (1 - \tau_k)^2\tau_{k-1}^2\left(\beta_k\hat{L}_i + \|A_i\|^2\right).$$

This condition holds if $\tau_k^2 = (1 - \tau_k)^2\tau_{k-1}^2$, which leads to $\tau_k = \frac{\tau_{k-1}}{\tau_{k-1}+1}$. This is the update rule (11) of the algorithm. It is trivial to show that $\tau_k = \frac{1}{k+\tau_0^{-1}}$ and $\beta_k = \frac{\beta_1}{\tau_0(k-1)+1}$. Now, we apply (44) to obtain the bound

$$S_{\beta_k}(\bar{x}^k) \leq \frac{C^*}{\tau_0(k - 1) + 1}, \quad \text{where } C^* := (1 - \tau_0)(F_{\beta_0}(x^0) - F^\star) + \sum_{i=1}^n\frac{\tau_0 B_i^0}{2q_i}\|x_i^\star - x_i^0\|_{(i)}^2.$$

Now, let us define the dual problem of (1) as

$$\max_{y \in \mathbb{R}^m} \left\{ \min_{x \in \mathbb{R}^p} F(x) + \langle Ax, y \rangle - h^*(y) \right\}, \tag{46}$$

and denote an optimal point of (46) as $y^\star$. We define $D_{\beta_k}(x) := F(x) + h_{\beta_k}(Ax) - F^\star$ and apply [9, Lemma 1] to obtain algorithm-independent duality bounds

$$\begin{cases} F(\bar{x}^k) - F^\star & \leq D_{\beta_k}(\bar{x}^k) + \|y^\star\| \, \|A\bar{x}^k - b\| + \frac{\beta_k}{2} \|y^\star - \dot{y}\|^2, \\ \|A\bar{x}^k - b\| & \leq \beta_k \left[ \|y^\star - \dot{y}\| + \left( \|y^\star - \dot{y}\|^2 + 2\beta_k^{-1} D_{\beta_k}(\bar{x}^k) \right)^{1/2} \right]. \end{cases} \tag{47}$$

The result in (12) follows by taking the expectation and using the concavity of the square-root and Jensen's inequality. $\qquad \square$

### B.3 The proof of Corollary 3.5

From the update in (13), we get the trivial inequality, similar to (27), that

$$\frac{\tau_k B_i^k}{2q_i} \|\bar{\tilde{x}}_i^{k+1} - \tilde{x}_i^k\|_{(i)}^2 \leq \langle \nabla_i \psi_{\beta_{k+1}}(\hat{x}^k), x_i^\star - \bar{\tilde{x}}_i^{k+1} \rangle$$
$$+ \frac{\tau_k B_i^k}{2q_i} \left( \|\bar{\tilde{x}}_i^{k+1} - x_i^\star\|_{(i)}^2 - \|\tilde{x}_i^k - x_i^\star\|_{(i)}^2 \right). \tag{48}$$

Due to the specific Step 8 in Section 3.5, instead of (33), we get

$$\mathbb{E}_{i_k} \left[ \hat{F}_{\beta_{k+1}}^{k+1} \mid \mathcal{F}_k \right] = \mathbb{E}_{i_k} \left[ f(\bar{x}^{k+1}) \mid \mathcal{F}_k \right] + \mathbb{E}_{i_k} \left[ \hat{g}^{k+1} \mid \mathcal{F}_k \right] + \mathbb{E}_{i_k} \left[ \varphi_{\beta_{k+1}}(\bar{x}^{k+1}) \mid \mathcal{F}_k \right]$$
$$\leq \left[ f(\hat{x}^k) + \tau_k \sum_{i=1}^n \langle \nabla_i f(\hat{x}^k), \bar{\tilde{x}}_i^{k+1} - \tilde{x}_i^k \rangle \right]$$
$$+ \left[ \varphi_{\beta_{k+1}}(\hat{x}^k) + \tau_k \sum_{i=1}^n \langle \nabla_i \varphi_{\beta_{k+1}}(\hat{x}^k), \bar{\tilde{x}}_i^{k+1} - \tilde{x}_i^k \rangle \right]$$
$$+ \sum_{i=1}^n \frac{\tau_k^2}{2q_i} \left( \hat{L}_i + \frac{\|A_i\|^2}{\beta_{k+1}} \right) \|\bar{\tilde{x}}_i^{k+1} - \tilde{x}_i^k\|_{(i)}^2. \tag{49}$$

Plugging (48) into the last inequality gives us

$$\mathbb{E}_{i_k} \left[ \hat{F}_{\beta_{k+1}}^{k+1} \mid \mathcal{F}_k \right] = \mathbb{E}_{i_k} \left[ f(\bar{x}^{k+1}) \mid \mathcal{F}_k \right] + \mathbb{E}_{i_k} \left[ \hat{g}^{k+1} \mid \mathcal{F}_k \right] + \mathbb{E}_{i_k} \left[ \varphi_{\beta_{k+1}}(\bar{x}^{k+1}) \mid \mathcal{F}_k \right]$$
$$\leq \left[ f(\hat{x}^k) + \tau_k \langle \nabla f(\hat{x}^k), x^\star - \tilde{x}^k \rangle \right] + \left[ \varphi_{\beta_{k+1}}(\hat{x}^k) + \tau_k \langle \nabla \varphi_{\beta_{k+1}}(\hat{x}^k), x^\star - \tilde{x}^k \rangle \right]$$
$$+ \sum_{i=1}^n \frac{\tau_k^2 B_i^k}{2q_i} \left( \|x_i^\star - \tilde{x}_i^k\|_{(i)}^2 - \|x_i^\star - \bar{\tilde{x}}_i^{k+1}\|_{(i)}^2 \right). \tag{50}$$

If we let $Q_k := \sum_{i=1}^n \frac{\tau_k^2 B_i^k}{2q_i} \left( \|x_i^\star - \tilde{x}_i^k\|_{(i)}^2 - \|x_i^\star - \bar{\tilde{x}}_i^{k+1}\|_{(i)}^2 \right)$, then similar to (40), we get

$$Q_k = \mathbb{E}_{i_k} \left[ \sum_{i=1}^n \frac{\tau_k^2 B_i^k}{2q_i^2} \left( \|x_i^\star - \tilde{x}_i^k\|_{(i)}^2 - \|x_i^\star - \tilde{x}_i^{k+1}\|_{(i)}^2 \right) \mid \mathcal{F}_k \right]. \tag{51}$$

Consequently, by using the same updates for $\tau_k$ and $\beta_k$, the recursion in (41) becomes

$$\mathbb{E} \left[ \hat{F}_{\beta_{k+1}}^{k+1} - F^\star \right] + \mathbb{E} \left[ \sum_{i=1}^n \frac{\tau_k^2 B_i^k}{2q_i^2} \|x_i^\star - \tilde{x}_i^{k+1}\|_{(i)}^2 \right] \leq (1 - \tau_k) \mathbb{E} \left[ \hat{F}_{\beta_k}^k - F^\star \right]$$
$$+ \mathbb{E} \left[ \sum_{i=1}^n \frac{\tau_k^2 B_i^k}{2q_i^2} \|x_i^\star - \tilde{x}_i^k\|_{(i)}^2 \right]. \tag{52}$$

Hence, we finally get

$$\mathbb{E}\left[F_{\beta_k}(\bar{x}^k) - F^\star\right] \le \frac{1}{\tau_0(k-1)+1}\left[(1-\tau_0)(F_{\beta_0}(x^0) - F^\star) + \sum_{i=1}^{n}\frac{\tau_0^2 B_i^0}{2q_i^2}\|x_i^\star - x_i^0\|_{(i)}^2\right].$$

Noting that $\tau_0 = 1$, we get

$$S_{\beta_k}(\bar{x}^k) \le \frac{C^*}{k}, \quad \text{where} \quad C^* := \sum_{i=1}^{n}\frac{B_i^0}{2q_i^2}\|x_i^\star - x_i^0\|_{(i)}^2. \tag{53}$$

By using the bound of $\{\beta_k\}_{k \ge 1}$ as in Lemma A.2, we obtain the bound (15). For the constrained case, we use (53) on (47), with the specific update rule of $\{\beta_k\}$ for the constrained case, to obtain (16) using the same arguments as in the Proof of Theorem 3.4. $\square$

## C  Equivalence of SMART-CD and Efficient SMART-CD

In this appendix, we give a proof by induction for the equivalence of Algorithm 1 and Algorithm 2 motivated by [4].

### C.1  The proof of Proposition 3.1

The claim trivially holds for $k = 0$ using the initialization of the parameters. Assume that the relations hold for some $k$. Using Step 7 of Algorithm 2, we have

$$\tilde{z}_{i_k}^{k+1} = \tilde{z}_{i_k}^k + t_{i_k}^{k+1}. \tag{54}$$

We can write from Step 6 of Algorithm 2 that

$$
\begin{aligned}
t_{i_k}^{k+1} &= \arg\min_{t \in \mathbb{R}^{p_{i_k}}} \left\{ \langle \nabla_{i_k} f(c_k u^k + \tilde{z}^k) + A_{i_k}^\top y_{\beta_{k+1}}^*(c_k A u^k + A\tilde{z}^k), t \rangle + g_{i_k}(t + \tilde{z}_{i_k}^k) \right.\\
&\qquad\qquad \left. + \frac{\tau_k B_{i_k}^k}{2\tau_0}\|t\|_{(i_k)}^2 \right\}\\
&= \arg\min_{t \in \mathbb{R}^{p_{i_k}}} \left\{ \langle \nabla_{i_k} f(\hat{z}^k) + A_{i_k}^\top y_{\beta_{k+1}}^*(A\hat{z}^k), t \rangle + g_{i_k}(t + \tilde{z}_{i_k}^k) + \frac{\tau_k B_{i_k}^k}{2\tau_0}\|t\|_{(i_k)}^2 \right\}\\
&= \arg\min_{t \in \mathbb{R}^{p_{i_k}}} \left\{ \langle \nabla_{i_k} f(\hat{x}^k) + A_{i_k}^\top y_{\beta_{k+1}}^*(A\hat{x}^k), t \rangle + g_{i_k}(t + \tilde{x}_{i_k}^k) + \frac{\tau_k B_{i_k}^k}{2\tau_0}\|t\|_{(i_k)}^2 \right\}\\
&= -\tilde{x}_{i_k}^k + \arg\min_{x \in \mathbb{R}^{p_{i_k}}} \left\{ \langle \nabla_{i_k} f(\hat{x}^k) + A_{i_k}^\top y_{\beta_{k+1}}^*(A\hat{x}^k), x - \hat{x}_{i_k}^k \rangle + g_{i_k}(x) \right.\\
&\qquad\qquad\qquad \left. + \frac{\tau_k B_{i_k}^k}{2\tau_0}\|x - \tilde{x}_{i_k}^k\|_{(i_k)}^2 \right\}\\
&= -\tilde{x}_{i_k}^k + \tilde{x}_{i_k}^{k+1}.
\end{aligned}
$$

By (54) and the inductive assumption on $\tilde{x}^k$, we obtain

$$\tilde{z}^{k+1} = \tilde{x}^{k+1}.$$

Next, using the definition of $\bar{z}^{k+1}$ and Step 8, we can derive

$$
\begin{aligned}
\bar{z}^{k+1} &= c_k u^{k+1} + \tilde{z}^{k+1} = c_k\left(u_k - \frac{1 - \tau_k/\tau_0}{c_k}(\tilde{z}^{k+1} - \tilde{z}^k)\right) + \tilde{z}^{k+1}\\
&= c_k u^k + \tilde{z}^k + \frac{\tau_k}{\tau_0}(\tilde{z}^{k+1} - \tilde{z}^k)\\
&= \hat{z}^k + \frac{\tau_k}{\tau_0}(\tilde{z}^{k+1} - \tilde{z}^k)\\
&= \hat{x}^k + \frac{\tau_k}{\tau_0}(\tilde{x}^{k+1} - \tilde{x}^k)\\
&= \bar{x}^{k+1}.
\end{aligned}
$$

Finally, we use the definition of $\hat{z}^{k+1}$, $c_k$ and Step 9 of Algorithm 1, we arrive at

$$
\begin{aligned}
\hat{z}^{k+1} &= c_{k+1} u^{k+1} + \tilde{z}^{k+1} \\
&= \frac{c_{k+1}}{c_k} (\bar{x}^{k+1} - \tilde{z}^{k+1}) + \tilde{z}^{k+1} \\
&= (1 - \tau_{k+1})(\bar{z}^{k+1} - \tilde{z}^{k+1}) + \tilde{z}^{k+1} \\
&= (1 - \tau_{k+1})(\bar{x}^{k+1} - \tilde{x}^{k+1}) + \tilde{x}^{k+1} \\
&= (1 - \tau_{k+1})\bar{x}^{k+1} + \tau_{k+1}\tilde{x}^{k+1} \\
&= \hat{x}^{k+1}.
\end{aligned}
$$

Hence, we can conclude that Algorithm 1 and Algorithm 2 are equivalent. $\qquad\square$