[Reviews · NeurIPS 2017]

Reviewer 1



This paper combines several techniques in large-scale optimization: smoothing, acceleration, homotopy, and non-uniform sampling. It solves the problem with three convex functions and a linear operator f(x)+g(x)+h(Ax), where f is smooth. The smoothing technique from Nesterov [14] is applied to smooth the last term h(Ax), then a method based on block-wise forward-backward splitting is used to update one randomly chosen block at every iteration. Then acceleration and homotopy is applied to obtain a faster convergent algorithm. In fact this algorithm is a primal algorithm using the proximal gradient on the smoothed function. The numerical results show the performance of the proposed algorithm compared with some existing algorithms. L165, the reference is missing.

Reviewer 2



The authors propose a new smooth primal-dual randomized coordinate descent method for the three composite problem F(x) = f(x) + g(x) + h(Ax), where f is smooth, g is non-smooth, separable and has a block-wise proximal operator, and h is a general nonsmooth function. The method is basically a generalization of the accelerated, parallel, and proximal coordinate descent method of Fercoq and Richtarik, and the coordinate descent method with arbitrary sampling of Qu and Richtarik to the three composite case. The difficulty of this extension lies in the non-smoothness of the function h composed with a linear operator A. The authors provide an efficient implementation by breaking up full vector updates. They show that the algorithm achieves the best known O(n/k) convergence rate. Restart schemes are also used to improve practical performance. Special cases of the three composite problem are considered and analyzed. The contributions are clearly stated and overall, this is a nice paper. It is applicable to many different objective functions and employs all the "tricks" to make this a fast and efficient method. Comments: - The smoothing and non-uniform sampling techniques of Algorithm 1 are clearly described in Section 2, but the homotopy and acceleration techniques are not described anywhere, they are just directly used in Alg 1. This makes Alg. 1 a little cryptic if you are unfamiliar with the structure of these methods -- I think Alg. 1 needs to be better explained either in words or indicating what method/approach each step corresponds to. - Following from the previous comment, ll 9 of Algorithm 1 - where does this formula for \tau come from?, ll 153 - where does the adapted \tau formula come from for the strongly convex case? This is not clear. - What are the values of \hat{x}, \tilde{x} and \bar{x} initialized to in Alg 1? - What is the "given" value of \dot{y} (re: line 73 and equation (6))? - Unknown reference in line 165. ============ POST REBUTTAL ============ I have read the author rebuttal and the other reviews. I thank the authors for addressing my comments/questions, as well as the comments/questions of the other reviewers. My recommendation stands, I think this paper should be accepted.

Reviewer 3



This paper combines several existing techniques to extend the applicability of coordinate descent method to a larger class of problem. I think this is a nice and solid work but just not has much novelty. The proof can be derived by modifying the proofs in the previous papers and the convergence rate is expected. I found the following issues which I wish the authors can fix. 1. (Major) The Assumption 1 (c) says h^* should have a bounded domain. However, this is not satisfied in (9). In fact, when h is an indicator function of a singleton set {c}, h^*(u)=u^Tc whose domain is R^n. Therefore, all results in Section 3.4, including Theorem 3.5, cannot be justified. I wonder if there is a mistake in the proof of Theorem 3.5. For example, how can D_h^* be finite when the domain is not bounded? If D_h^* is unbounded, how can the right hand side of (12) depends on D_h^*? (Minor) 2. I found both y^* and y^\star in Theorem 3.5. Please make the notation consistent. Also I did not find their definitions, although I can guess they are dual optimal solution. 3. There is a missing reference [?] in line 165.